# Adult-Onset CNS Sulfatide Deficiency Causes Sex-Dependent Metabolic Disruption in Aging

**DOI:** 10.3390/ijms241310483

**Published:** 2023-06-22

**Authors:** Shulan Qiu, Sijia He, Jianing Wang, Hu Wang, Anindita Bhattacharjee, Xin Li, Moawiz Saeed, Jeffrey L. Dupree, Xianlin Han

**Affiliations:** 1Barshop Institute for Longevity and Aging Studies, University of Texas Health Science Center at San Antonio, San Antonio, TX 78229, USA; shulanqiu1@gmail.com (S.Q.); hes3@uthscsa.edu (S.H.);; 2Division of Diabetes, Department of Medicine, University of Texas Health Science Center at San Antonio, San Antonio, TX 78229, USA; 3Department of Anatomy and Neurobiology, Virginia Commonwealth University, Richmond, VA 23284, USA; 4McGuire Veterans Affairs Medical Center, Research Division, Richmond, VA 23249, USA

**Keywords:** sulfatide, glucose metabolism, food intake, Alzheimer’s disease, aging

## Abstract

The interconnection between obesity and central nervous system (CNS) neurological dysfunction has been widely appreciated. Accumulating evidence demonstrates that obesity is a risk factor for CNS neuroinflammation and cognitive impairment. However, the extent to which CNS disruption influences peripheral metabolism remains to be elucidated. We previously reported that myelin-enriched sulfatide loss leads to CNS neuroinflammation and cognitive decline. In this study, we further investigated the impact of CNS sulfatide deficiency on peripheral metabolism while considering sex- and age-specific effects. We found that female sulfatide-deficient mice gained significantly more body weight, exhibited higher basal glucose levels, and were glucose-intolerant during glucose-tolerance test (GTT) compared to age-matched controls under a normal diet, whereas male sulfatide-deficient mice only displayed glucose intolerance at a much older age compared to female sulfatide-deficient mice. Mechanistically, we found that increased body weight was associated with increased food intake and elevated neuroinflammation, especially in the hypothalamus, in a sex-specific manner. Our results suggest that CNS sulfatide deficiency leads to sex-specific alterations in energy homeostasis via dysregulated hypothalamic control of food intake.

## 1. Introduction

Increased body weight and obesity pose a major global health risk which increases the prevalence of multiple associated diseases including diabetes, cardiovascular diseases, and neurodegenerative diseases. Elevated body weight and obesity are a very complex and multifactorial problem characterized by energy imbalance. This imbalance is linked to environmental conditions, genetic factors, and energy expenditure. Over the past decades, numerous studies have supported the role of the CNS in regulating glucose levels. The CNS communicates with the liver and other peripheral organs to regulate blood glucose levels via specific neurons, circulating hormones, and specific nutrients. Both insulin-dependent and -independent processes contribute to fasting and postprandial plasma glucose regulation, and the nervous system involves these processes in both direct and indirect manners [1]. An increasing number of studies suggest that there is a reciprocal cause-and-effect relationship between neuroinflammation and elevated body weight and obesity. Excessive body weight and obesity can lead to neuroinflammation [2,3]. Conversely, neuroinflammation, which is characterized by an overactive immune response in the brain, has been linked to several disorders, including obesity. Abundant evidence shows that central inflammation, especially hypothalamic inflammation, plays a key role in the association among leptin/insulin resistance, obesity, and peripheral metabolic dysfunction under a high-fat (HF) diet [4,5,6].

The brain tissue has high lipid content and diversity, mainly due to the abundance of lipid-enriched myelin [7]. Sulfatide is a major lipid component in the nervous system. It is found in high levels on the extracellular leaflet of the myelin sheath produced by oligodendrocytes and Schwann cells in the CNS and peripheral nervous system (PNS), respectively. Sulfatide is a class of sulfoglycolipids, the sulfate of which is transferred by a glycolipid-specific sulfotransferase, i.e., cerebroside sulfotransferase (CST, which is coded by the *Gal3st1* gene) [8,9]. We and many other laboratories have revealed that brain sulfatide content is specifically and dramatically reduced at the earliest clinically recognizable stages of Alzheimer’s disease (AD), including in gray matter, white matter, and cerebrospinal fluid [10,11,12,13,14,15,16,17,18,19,20,21]. Likewise, significant losses of brain sulfatide have been reported in multiple AD mouse models [11,12,13,14,15,16]. Using an adult-onset sulfatide deficiency mouse model, our previous studies detected a marked induction of neuroinflammation in the CNS after loss of myelin sulfatides [22]. Given the profound regulatory effect of the CNS on peripherical energy metabolism, whether CNS sulfatide deficiency and the related neuroinflammation affect the metabolism and body weight during aging remains to be explored.

Another important unanswered question is why increased rates of AD occur in women, as pointed out by sufficient biological evidence. Women represent over 65% of the cases of late-onset AD. Current research has focused on differential risks [23,24], including *Apoe4* [25], to explain the higher incidence of AD in women. Interestingly, obesity also appears to be more prevalent among women than men. However, the related molecular/cell mechanisms underlying the gender difference in AD and associated peripheral metabolic disorders remain entirely unclear.

In this study, using adult-onset sulfatide-deficient mice, a mouse model with inducible and conditional depletion of the CST gene, we found that adult-onset sulfatide deficiency resulted in marked metabolic disruption, including increased body weight, glucose intolerance, and neuroinflammation, which are all well-known AD risk factors. In addition, it is surprising that weight, glucose intolerance, and neuroinflammation in adult-onset sulfatide-deficient mice all showed sex-dependent differences, which is further supported by transcriptomic analysis showing sex dimorphic induction of immune/inflammation pathways. In conclusion, our study reveals, for the first time, that deficiency in a class of myelin-specific lipids causes a sex-specific abnormality in glucose metabolism and excessive body weight gain. Our findings provide valuable information to help better understand the interconnection between CNS myelin lipid homeostasis and peripheral energy metabolism with an emphasis on sex-specific responses.

## 2. Results

### 2.1. CNS Myelin Sulfatide Depletion Induces Excessive Body Weight Gain and Impairs Glucose Tolerance with Gender Differences under a Normal Diet

Conditional CST knockout (CST cKO) mice and their respective control mice were injected with tamoxifen at the age of 4 mo to induce oligodendrocyte-specific knockout of CST (*Gal3st1* gene) driven by Plp1-CreERT. Surprisingly, even under a normal diet, CST cKO mice gained significantly higher body weight six mo after tamoxifen injection compared with their respective controls. Furthermore, only female, but not male, CST cKO mice gained significantly higher weight, which was confirmed by two independent cohorts (Figure 1A). To test if the age of CST knockout affects the sex dimorphism in body weight gain, we injected tamoxifen in a separate cohort of 12 mo old mice. Male CST cKO mice gained significantly more body weight 6 mo post tamoxifen injection compared with the control mice (Figure 1B), which suggests that the effect of CST cKO on regulating body weight follows a sex-specific manner with a potential impact from age of sulfatide loss. To further evaluate the source of increased body weight, we performed a qMRI test to measure the body composition of cKO and control mice. We found a prominent increase in fat accumulation with no change in lean or water weight from CST cKO mice vs. controls (Figure 1C). This finding indicates that the impact of myelin lipid loss on body weight is potentially due to disrupted energy metabolism.

To further investigate whether there is any impact of CST cKO on peripheral organ metabolism, we measured blood glucose levels and found elevated blood glucose in CST cKO mice without (Appendix A) and with 4 h fasting (Figure 2A) compared with controls, respectively, under a normal diet. Interestingly, we observed increased glucose levels in female mice much earlier than male mice upon inducing sulfatide deficiency, evidenced by the observation that higher glucose levels were detected in female sulfatide-deficient mice at 6 mo after tamoxifen injection, while the same phenotype was detected in male mice at 15 mo post-injection. Given the elevated glucose levels in cKO mice, we hypothesized that glucose metabolism may be disturbed due to the sulfatide loss. We performed a glucose tolerance test (GTT) and found that female, but not male, CST cKO mice showed impaired glucose tolerance compared to their controls (Figure 2B). To further pinpoint whether the impaired glucose management was due to altered insulin-stimulated glucose uptake, we performed an insulin tolerance test (ITT). Interestingly, no difference was detected during ITT in both sexes (CST cKO vs. controls) (Figure 2C). In addition, no change was detected in plasma insulin levels (Figure 2D) upon CST knockout. This suggests that the pancreatic insulin secretion as well as insulin responsiveness of insulin-sensitive metabolic tissues are largely preserved in CNS sulfatide-deficient mice under a normal diet.

### 2.2. CNS Myelin Sulfatide Depletion Causes Increased Food Intake in Female Mice

Obesity or increased body weight suggests a positive energy balance, which usually results from an imbalance between energy intake and expenditure [26]. Increased food intake and decreased exercise causes a positive energy balance [27]. To determine the causal reason for increased body weight gain in CST cKO mice, we evaluated the activity and food intake of these mice. Interestingly, mice 9 mo post tamoxifen injection did not present any changes in spontaneous locomotor activity (Figure 3A), while analysis of food consumption revealed increased food intake in CST cKO female mice (Figure 3B), which may be a contributing factor in the elevated glucose levels and body weight gain. It is reported that food intake can be regulated by the CNS through leptin signaling; however, we detected no significant change in plasma leptin levels between cKO and controls (Figure 3C), suggesting that the regulation of elevated food intake in female cKO mice is potentially mediated in a leptin-independent manner.

### 2.3. CNS Myelin Sulfatide Depletion Induces a Chronic Immune/Inflammatory Response, Including Hypothalamic-Inflammation-Related Pathway Signaling with Gender Differences

We previously reported that sulfatide deficiency caused neuroinflammation in the CNS [22]. To further investigate whether sulfatide deficiency also leads to whole-body inflammation, which might be related to the body weight gain, we tested plasma levels of 23 cytokines and chemokines in 9 and 12 mo post tamoxifen injection mice. None showed significant changes, indicating the absence of peripheral inflammation. It has been reported that inflammation in the hypothalamus, the part of the brain responsible for regulating appetite and metabolism, can lead to an increased appetite and decreased energy expenditure, resulting in weight gain [28,29,30], a condition known as “hypothalamic obesity”. Based on this, we investigated the association between hypothalamic inflammation and body weight gain in CST cKO mice. Cerebral immunostaining revealed the activation of GFAP-positive astrocytes in the inner cortical layers, corpus callosum (CC), and the partial hippocampal substructure of cCK mice compared to controls (Figure 4A), and the extent of activation appears to be more significant in females versus males. The different degree of glial activation between genders was further confirmed by the observation of Iba1-positive microglia and GFAP-positive astrocytes in the hypothalamus (Figure 4B).

To understand the impact of sulfatide loss on brain alterations on a transcriptional level, we used the Mouse Neuroinflammation Panel from NanoString nCounter^®^ Technology, which consists of 770 genes that represent 22 different pathways primarily related to immune response/inflammation, to further explore the detailed mechanisms underlying the sex-related differences caused by sulfatide deficiency. Our previous results showed that sulfatide deficiency caused neuroinflammation in both the cerebrum and spinal cord. The neuroinflammation seemed to show up earlier in the spinal cord than in the cerebrum, possibly because the spinal cord is more enriched with sulfatide [22]. Thus, we focused our analysis of the sex-related differences in the mechanisms of neuroinflammation using spinal cord tissue collected from male and female CST cKO mice, 6 mo post tamoxifen treatment. Principal component analysis (PCA) showed that female CST Cre^+^ groups were better separated from the corresponding CST Cre^−^ control samples than those from male mice (Appendix A). Higher numbers of DEGs (CST Cre^+^ vs. CST Cre^−^) were found in female mice (122 DEGs) than in male mice (53 DEGs), from which only 12 DEGs overlapped between sexes (Figure 5A and Appendix A). In addition, the majority of the upregulated DEGs (indicated as orange dots in the red-lined rectangle in the volcano plot displaying −log10 (*p*-value) and log2 (fold change) of genes in Figure 5B) were related to microglia/astrocyte function and inflammatory signaling. Specifically (as shown in Figure 5C,D), markers for microglial activation (Cd68, Trem2, and C1qa) and astrocyte activation (Vim, Serpina3n, and Osmr) were regulated in a sex-specific manner after CST knockout. Furthermore, the sulfatide deficiency also resulted in the upregulation of Apoe, an important gene implicated in AD, in a sex-dependent manner (Figure 5E). Hypothalamic inflammation is a complex process, and several cellular signaling pathways have been identified in its pathogenesis [31,32,33,34]. KEGG pathway enrichment analysis of the 110 female-specific DEGs revealed that several hypothalamic-inflammation-related signaling pathways were significantly altered after sulfatide loss (Figure 5F), including the MAPK signaling, Nfκb signaling, and TLR4 pathway, and so on.

## 3. Discussion

The relationship among increased body weight and obesity, aging, neuroinflammation, dementia, and gender differences is intricate and multifactorial. Increased body weight and obesity are associated with numerous adverse health outcomes, including an increased risk of dementia [26]. Research has shown that being overweight or obese in mid-life may heighten the risk of developing dementia later in life [35,36]. On the other hand, AD is the most prevalent cause of dementia in aging [37,38], and AD has a possible link with hyperglycemia, which was confirmed in previous studies [39,40,41]. While AD is characterized by the presence of extracellular Aβ plaques and intracellular hyperphosphorylated tau neurofibrillary tangles, increasing evidence has implicated sustained glia-mediated inflammation as a major contributor to AD neurodegenerative processes and cognitive deficits (reviewed in [42,43]). Neuroinflammation, characterized by the activation of glial cells in the CNS, has been confirmed as a risk factor of AD [44]. Interestingly, neuroinflammation has also been implicated in the development of obesity [45]. Chronic inflammation can lead to alterations in the hypothalamic–pituitary–adrenal (HPA) axis, culminating in increased appetite and decreased energy expenditure [4,46,47,48]. Consequently, obesity and neuroinflammation (which are both associated with AD) may share a reciprocal cause-and-effect relationship.

Another aspect of some pathologies is the sex difference. The World Health Organization (WHO) reports that global prevalence of obesity is slightly higher among women than men [49]. However, the distribution of obesity may vary significantly, depending on the region, culture, and socioeconomic status. Interestingly, women are more likely to develop AD than men, accounting for approximately two-thirds of all cases [24,50]. Although the molecular mechanisms underlying gender differences in AD remain poorly understood, differences in the immune system between men and women have been proposed as a contributing factor.

Our lab and others have established that disrupted lipid metabolism is present in AD pathogenesis and it functions as an important factor to induce neuroinflammation [15,17,22]. Sulfatide, a class of sphingolipids abundant in the brain’s myelin sheath, plays a crucial role in the nervous system. Previous studies have reported significant losses of brain sulfatide content in early preclinical stages of AD in humans [19] and in animal models, with sulfatide loss exacerbating with age [11,12,13,14,15,16]. Our recent studies using an adult-onset sulfatide deficiency mouse model found that CNS sulfatide loss in myelinating cells is sufficient to activate disease-associated microglia and astrocytes, leading to chronic AD-like neuroinflammation and cognitive impairment [22]. We also demonstrated that adult-onset sulfatide deficiency led to a progressive loss of axonal protein domain organization and brain ventricular enlargement [51,52]. Considering the significant changes observed in the CNS, including the induction of neuroinflammation, we hypothesized that sulfatide loss might also have an impact on the metabolism of peripheral organs. This study aimed to investigate the effects of CNS myelin sulfatide depletion on body weight, glucose intolerance, energy homeostasis, and inflammation, with a focus on gender differences.

Our results demonstrated that, even under a normal diet, CST cKO mice gained significantly more body weight at 10 mo of age (6 mo post tamoxifen induction) compared to their respective control mice. Interestingly, this increase in body weight was observed only in female CST cKO mice. However, when tamoxifen was injected in 12 mo old male CST cKO mice, they gained significantly more body weight than controls at 6 mo post injection, suggesting a sex dimorphic mechanism of body weight regulation in sulfatide deficiency. Our findings also indicated that increased fat accumulation was the major contributing factor for body weight gain in CST cKO mice, as evidenced by qMRI test results. In parallel, we observed increased blood glucose levels in CST cKO mice compared to the respective control groups, which also occurred earlier in female mice than in male mice after induced sulfatide deficiency. Consistent with the blood glucose measurements, female CST cKO mice were less tolerant to glucose than their control counterparts at around 12 mo old according to GTT.

We previously reported that sulfatide deficiency led to AD-like neuroinflammation [22]. Although we did not find significant changes in cytokines or chemokines in plasma to indicate systemic inflammation, our current study further investigated the association between hypothalamic inflammation and increased body weight in CST cKO mice in a sex-dependent manner. Immunostaining of Iba1 and GFAP in cerebrum, including the hypothalamus, revealed active microglia and astrocytes in CST cKO mouse brains. Notably, gender differences in inflammation were observed in the hypothalamus, inner cortical layers, corpus callosum, and some hippocampus substructures. Additionally, the NanoString nCounter^®^ Mouse Neuroinflammation Panel also clearly revealed sex-dependent differences in microglia/astrocyte activation and inflammatory signaling. Hypothalamic inflammation is a complex process, and several cellular signaling pathways have been identified to play a role in its pathogenesis, including the MAPK, NF-κB, TLR4, and many other pathways [31,32,33,34]. These signaling pathways were also confirmed to activate in a sex-dimorphic manner in the CST cKO mice. All these results correspond with increased food intake, which may contribute to the observed increase in blood glucose and body weight in female CST cKO mice.

In conclusion, our study suggests that CNS myelin sulfatide depletion leads to increased body weight, impaired glucose tolerance, and inflammation, with notable pathogenic differences influenced by sex (Figure 6). Our findings contribute to the understanding of the complex interplay among CNS myelin sulfatide depletion, energy homeostasis, and gender-specific responses. Further research is necessary to elucidate the molecular mechanisms underlying these observations and develop potential therapeutic strategies for managing the effects of sulfatide deficiency.

### Limitations

The current study utilized a mouse model of inducible sulfatide deficiency in myelinating cells, which has been reported to initiate neuroinflammation in multiple regions of the brain. It remains unclear whether the disruption of glucose metabolism and the dysregulation of body weight gain are solely attributed to neuroinflammation in the hypothalamus. Furthermore, future investigation is required to identify the precise mechanistic factors contributing to the sex-related differences in metabolic disruption observed in conditions of sulfatide deficiency.

## 4. Materials and Methods

### 4.1. Mice

The CST loxP/loxP (CSTfl/fl) mouse model was generated and used as described in our previous study [22]. Briefly, it was created by using the Clustered Regularly Interspaced Short Palindromic Repeats (CRISPR) technology. Then, CSTfl/fl mice were crossed with Plp1-CreERT+ mice (Stock No: 005975, the Jackson Laboratory, Bar Harbor, ME, USA). CST conditional knockout mice (CSTfl/fl/Plp1-CreERT− (CST Cre−) and CSTfl/fl/Plp1-CreERT+ (CST Cre+, CST cKO)) were treated with tamoxifen in order to induce CreERT-mediated genomic recombination for excision of exon 3 and exon 4 of the *Gal3st1* gene. Tamoxifen was injected intraperitoneally at a dosage of 40–60 mg/kg body weight once every 24 h for a total of 4 consecutive days. All the mice were housed in groups of ≤5 mice/cage, and the protocols for animal experiments were conducted in accordance with the “Guide for the Care and Use of Laboratory Animals” (8th edition, National Research Council of the National Academies, 2011) and were approved by the Animal Studies Committee of The University of Texas Health Science Center at San Antonio (Protocol 20180044AP, approval date: 9 January 2019).

### 4.2. Animal Behavior

The measurement of 48 h activity was performed by the Integrated Physiology of Aging Core of San Antonio Nathan Shock Center.

### 4.3. Brain Preparation

For histological analysis, mice were anesthetized with isoflurane and perfused with PBS. Right-brain hemispheres were fixed in 4% PFA overnight and placed in 10%, 20%, and 30% sucrose solution subsequently before freezing in optimal cutting temperature (OCT) compound, then were cut on a freezing sliding microtome. Serial 10 µm coronal sections of the brain were collected. For protein and mRNA expression analyses, the left-brain hemispheres were dissected out and flash-frozen in liquid nitrogen.

### 4.4. Gene Expression Analysis

Brain tissue was frozen in liquid nitrogen and powdered. RNA was extracted by using the Animal Tissue RNA Purification Kit (Norgen, Thorold, ON, Canada), then the concentration of RNA was determined. Gene expression profiling analysis was performed using the NanoString nCounter^®^ Technology with the Mouse Neuroinflammation Panel and nCounter^®^ SPRINT™ Profiler (NanoString Technologies, Seattle, WA, USA) according to the manufacturer protocol. The data were analyzed using nSolver 4.0 software. While analyzing the data, the background was subtracted using the mean of negative controls, standard normalization was performed with positive control normalization and code set content normalization. All gene expression levels follow normal distribution according to Shapiro–Wilk normality test. The following genes were used as housekeeping genes for analysis between samples: *Supt7l, Lars, Tada2b, Csnk2a2, Aars, Xpnpep1*, and *Ccdc127*.

### 4.5. Immunofluorescence Staining

Goat serum (10%, Sigma, St. Louis, MO, USA) was used to block the frozen slice for 1 h at room temperature, then sections were incubated with anti-GFAP (chicken, Millipore, Burlington, MA, USA; rabbit, Dako, Japan), and anti-Iba1 (rabbit, FUJIFILM Wako Pure Chemical Corporation, USA) primary antibodies at 4 °C overnight, and the fluorescence-labeled second antibody (Invitrogen, Waltham, MA, USA) was incubated for 1 h at room temperature; then mounted with DAPI after three wash cycles. Images were captured with a confocal laser-scanning microscope (Zeiss LSM710, Germany).

### 4.6. Statistics

Data in figures are presented as mean ± SEM. All statistical analyses were performed using Prism (GraphPad). Two-way ANOVA with Bonferroni post hoc test for multiple comparisons was used to compare for multiple groups. Comparisons of two groups were performed using a two-tailed unpaired *t*-test. * *p* < 0.05, ** *p* < 0.01, and *** *p* < 0.001.

## Figures and Tables

**Figure 1 ijms-24-10483-f001:**
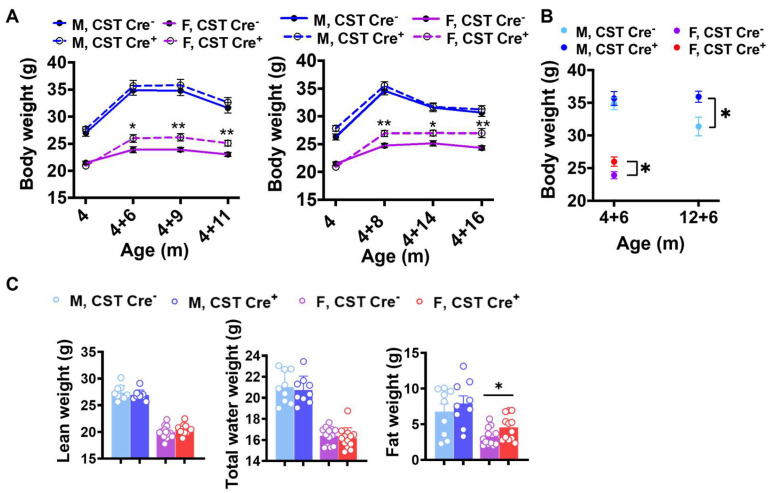
Age-related weight gain in CST cKO mice. (**A**) Body weight evolution of CST Cre− and CST Cre+ mice under normal diet from two independent cohorts (*n* = 9/genotype for male, *n* = 13–14/genotype for female). (**B**) Body weight at 6 mo post tamoxifen injection but with different injection age (*n* = 6–11/genotype for male, *n* = 5–7/genotype for female). (**C**) qMRI results from mice 11 mo post tamoxifen injection (*n* = 9/genotype for male, *n* = 13/genotype for female). Multiple *t*-test. * *p* < 0.05, ** *p* < 0.01.

**Figure 2 ijms-24-10483-f002:**
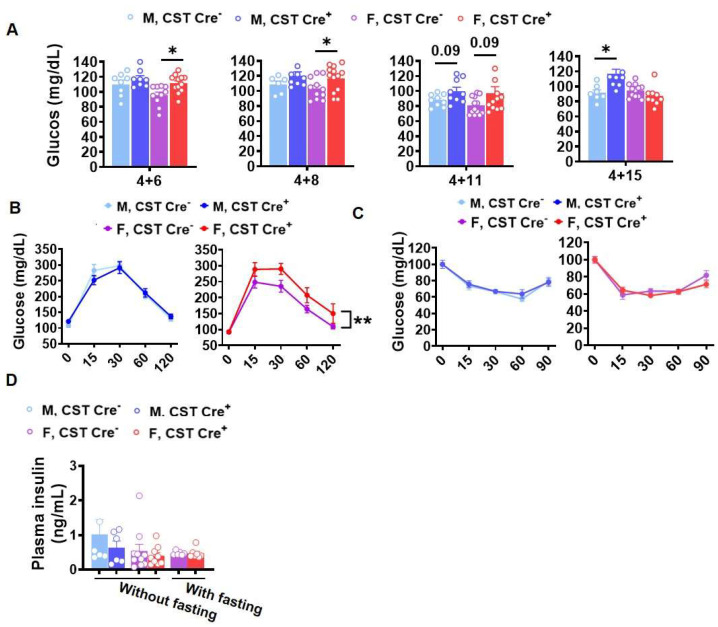
Sex- and age-specific increase of blood glucose levels in CST cKO mice. (**A**) Blood glucose levels measured after a 4 h fasting period at 10, 12, 15, and 19 mo of age under a normal diet. All groups were given a tamoxifen treatment at 4 mo (*n* = 6–9/genotype for male, *n* = 10–14/genotype for female). (**B**) Glucose tolerance test (GTT) after 8 h fasting at 15 mo (4 mo TAM +11) under a normal diet (*n* = 9/genotype for male, *n* = 13/genotype for female). (**C**) Insulin tolerance test (ITT) in 15 mo (4 mo TAM +11) old mice after a 4 h fasting period (*n* = 8/genotype for male, *n* = 12/genotype for female). (**D**) Plasma insulin levels with and without 4 h fasting measured by ELISA (*n* = 6/genotype for male, *n* = 7–10/genotype for female). (**A**,**D**) Multiple *t*-test. (**B**,**C**) Two-way ANOVA. * *p* < 0.05, ** *p* < 0.01.

**Figure 3 ijms-24-10483-f003:**
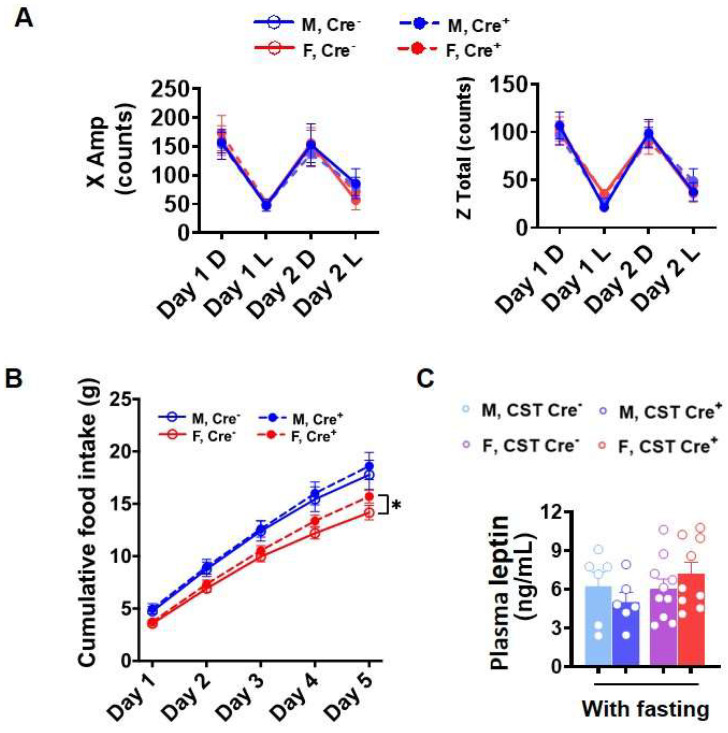
Food intake and 48 h activity. (**A**) Spontaneous locomotor activity from 48 h activity test (D: dark, L: light) (*n* = 9/genotype for male, *n* = 13/genotype for female). (**B**) Cumulative food intake for 5 days (*n* = 6–7/genotype for male, *n* = 8–9/genotype for female). (**C**) Plasma leptin with or without 4 h fasting measured by ELISA (*n* = 6/genotype for male, *n* = 9–10/genotype for female). (**A**) Two-way ANOVA. (**B**,**C**) Multiple *t*-test. * *p* < 0.05.

**Figure 4 ijms-24-10483-f004:**
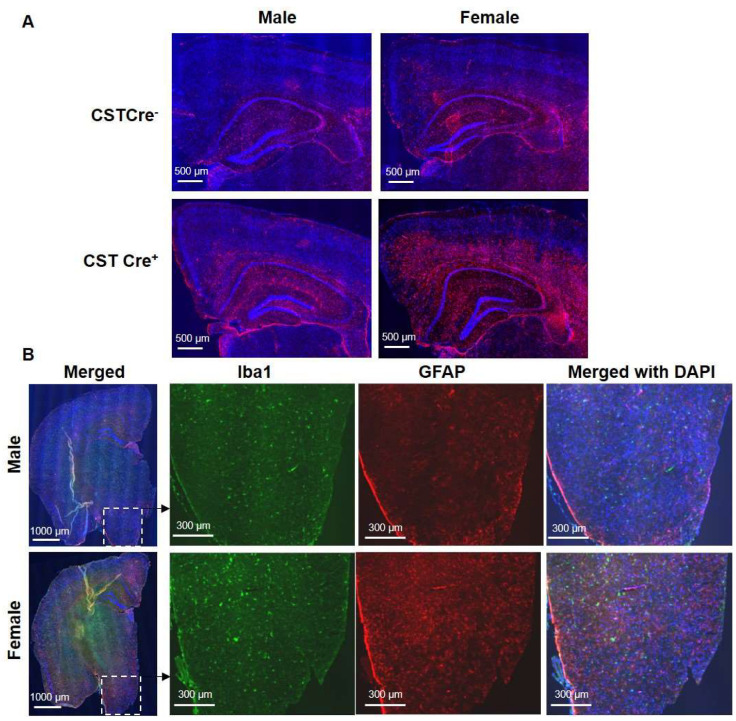
Increased hypothalamic inflammation in female, but not male, CST cKO mice at 13 months of age. (**A**) GFAP IF staining shows the sex-dependent astrocyte activation in inner cortical layers, corpus callosum (CC), and partial hippocampal substructure. (**B**) The immunostaining of cerebral Iba1 and GFAP in male and female CST cKO mice, with magnified images of the hypothalamus (*n* = 5/genotype/sex).

**Figure 5 ijms-24-10483-f005:**
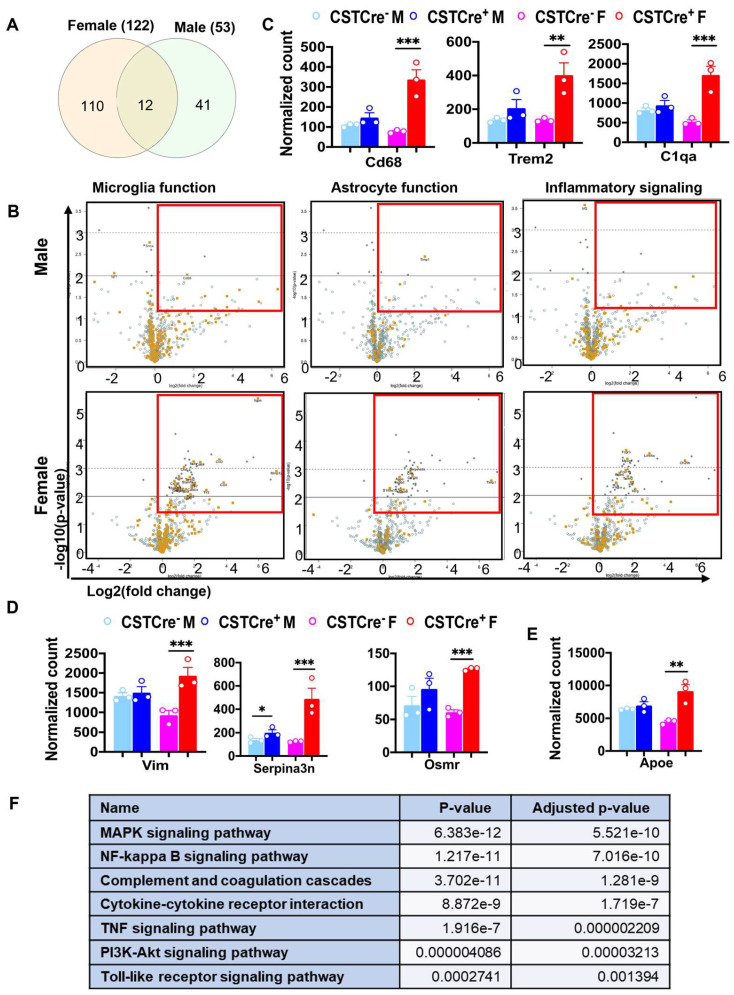
Myelin sulfatide deficiency induced neuroinflammation with sex difference. (**A**) Venn diagrams showing the number of specific and shared upregulated DEGs from female and male CST Cre+ vs. CST Cre−. Other DEGs Listed in Appendix A. (**B**) Volcano plot displaying −log10 (*p*-value) and log2 (fold change) for microglia or astrocyte function genes and inflammatory signaling from female or male CST Cre+ vs. CST Cre−. Orange dots indicate the respective function-related genes and gray dots indicate genes for other function in the panel. Red rectangles indicate the area with significant upregulated DEGs. (**C**) Counts for DEGs of specific markers of microglia activation (Cd68, Trem2 and C1qa). (**D**) Counts for DEGs of specific markers of astrocyte activation (Vim, Serpina3n and Osmr). (**E**) Counts for DEG of Apoe. (**F**) KEGG analysis for hypothalamic-inflammation related pathways using the specific 110 DEGs in female CST Cre+ vs. CST Cre− using the KEGG analysis tool on Enrichr. (**C**–**E**) Heteroscedastic Welch’s *t*-test, *n* = 3/genotype for male and female, respectively. * *p* < 0.05, ** *p* < 0.01, *** *p* < 0.001.

**Figure 6 ijms-24-10483-f006:**
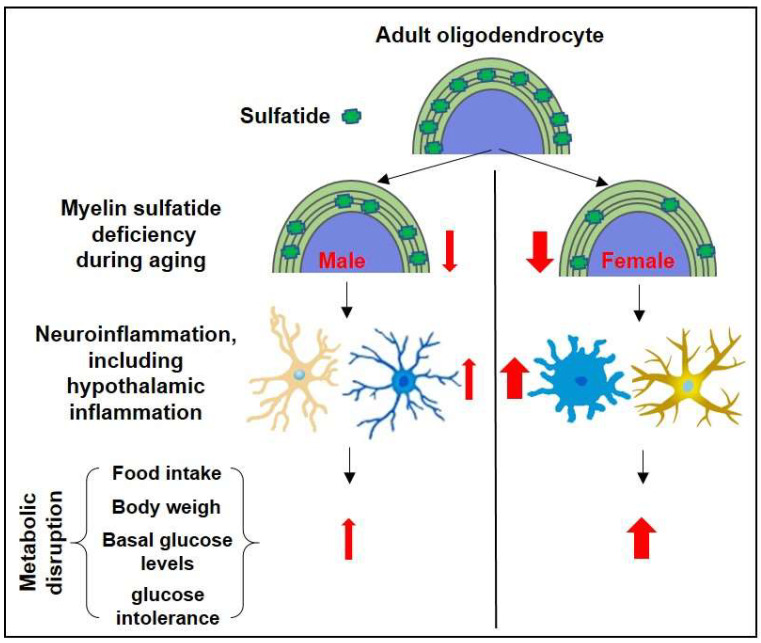
The schematic summary diagram showing the relationship between sulfatide loss, neuroinflammation (including hypothalamic inflammation), and the sulfatide deficiency-induced metabolic disruption.

## Data Availability

The data presented in this study are available within the article and Appendix A.

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
