# Peer review of "Adult-Onset CNS Sulfatide Deficiency Causes Sex-Dependent Metabolic Disruption in Aging"

_ijms, 2023, doi:10.3390/ijms241310483_

Round 1
Reviewer 1 Report
SUMMARY OF THE STUDY
By recurring to a mouse model of adult-onset sulfatide deficiency, where the cerebroside sulfotransferase gene can be selectively depleted, Qiu et al., discovered significant disruptions in metabolism. These disruptions included increased body weight, glucose intolerance, and neuroinflammation, all of which are well-known risk factors for Alzheimer's disease. Interestingly, the authors observed sex-dependent differences in weight, glucose intolerance, and neuroinflammation in these sulfatide-deficient mice. These findings are further supported by transcriptomic analysis, which revealed sex-specific activation of immune and inflammation pathways. Thus, according to the authors, this study represents the first evidence that a deficiency in myelin-specific lipids can lead to sex-specific abnormalities in glucose metabolism and excessive weight gain. Overall, the results of this paper provide valuable insights into the connection between CNS myelin lipid regulation and peripheral energy metabolism, with a particular emphasis on sex-specific responses.
GENERAL COMMENTS
The manuscript is well-written and provides a comprehensive – still preliminary - overview of the complex role of CNS sulfatide deficiency in leading to sex-specific alterations in energy homeostasis via dysregulated hypothalamic control of food intake.
Overall, there are very few criticisms I can make of this paper. I have detailed my comments below.
The title properly reflects the subject of the paper.
The abstract provides an accessible summary of the manuscript.
The keywords accurately reflect the content.
The introduction sets out the argument, summarizes recent research related to the topic, and highlights gaps in current understanding or conflicts in current knowledge.
The results are clearly presented and discussed.
The paper has an appropriate length.
The references are balanced, updated, and complete.
The supplementary material is clearly organized and well described.
SPECIFIC COMMENTS
What does appear to be missing is the rationale for treating mice with Tamoxifen (i.e., it induces cellular stress in the nervous system by inhibiting cholesterol synthesis). This is an important point to be introduced as not all the readers are familiar with that.
I would like to recommend the authors add a figure summarizing the study’s rounds. I would be very helpful to the readers.
Although the methods are appropriate, it is not clear to me what housekeeping genes have been used for the gene expression analyses. Moreover, I would like to ask the authors if their data were normally distributed and – if so – I recommend specifying that in the manuscript.
Authors should also justify the differences in the number of male and female mice used in the study.
I would also recommend the authors include study limitations in the discussion.
High quality of English Language
Author Response
Response to Reviewer 1
We thank the reviewers for his/her constructive comments. In response to reviewer 1’s overall comments, we have improved the method description and clarified the result presentation.
We have also included a point-to-point response below to reviewers’ specific comments:
- What does appear to be missing is the rationale for treating mice with Tamoxifen (i.e., it induces cellular stress in the nervous system by inhibiting cholesterol synthesis). This is an important point to be introduced as not all the readers are familiar with that.
Response: We appreciate reviewer for raising this point. We have updated in the results section and method description to include our rationale for treating mice with Tamoxifen (please see lines 94-96 and lines 435-437). To control for the potential adverse effects of tamoxifen treatment, we treated both Cre- and Cre+ groups in parallel with tamoxifen, thus the metabolic phenotype observed between Cre- and Cre+ are due to the loss of Sulfatides (as a result of Cre+ mediated Gal3st gene deletion).
- I would like to recommend the authors add a figure summarizing the study’s rounds. I would be very helpful to the readers.
Response: We appreciate reviewer for this suggestion. We generated a summary figure and included in our Figures section (Fig. 6) with a description in line 393.
- Although the methods are appropriate, it is not clear to me what housekeeping genes have been used for the gene expression analyses. Moreover, I would like to ask the authors if their data were normally distributed and – if so – I recommend specifying that in the manuscript.
Response: We thank reviewer for the comment. We have included the list of housekeeping genes used in NanoString analysis, which include Supt7l, Lars, Tada2b, Csnk2a2, Aars, Xpnpep1, and Ccdc127. All gene expression levels follow normal distribution according to Shapiro-Wilk normality test. We have specified both information in the method section of the manuscript (lines 462-466).
- Authors should also justify the differences in the number of male and female mice used in the study.
Response: We thank reviewer for this suggestion. We have updated the number of animals used for each sex respectively in all the figure legends. On average, we used n=6~11/genotype for male and n=5~13/genotype for female in phenotype evaluation studies, n=5/genotype/sex was used for histology, and n=5/genotype/sex for NanoString studies, which are commonly used in the literature.
- I would also recommend the authors include study limitations in the discussion.
Response: We appreciate reviewer for this comment. We have included a limitation section in the discussion of the manuscript. Please see lines 420-427.
We hope that we have adequately addressed the questions identified in the previously submitted manuscript and that the revised manuscript will meet the reviewers’ approval. Thank you for your encouragement in resubmitting this manuscript.
Sincerely,
Xianlin Han, PhD
Professor of Medicine
Barshop Institute for Longevity and Aging Studies
Department of Medicine
UTHealth San Antonio
Phone: (210) 562-4104
Email: hanx@uthscsa.edu
Reviewer 2 Report
After reading the manuscript in detail, I conclude that the abstract, introduction, and other chapters cover the issues discussed in an extensive and proper manner. The conclusions presented by the authors are related to the main research issue.
The weakness of the reviewed work concern as follows:
- The introduction seems to be too long. I recommend to transfer some parts to the discussion and compare literature data with results of the study in the discussion part. For example in lines 62-75 authors described own studies which correspond to the experiment from the manuscript
- The small number of citations of works from the last 5 years is rather surprising (out of 50 bibliographic items, only 20 works have been published since 2018). Authors need to conduct detailed survey of the literature and supplement the citation with references to recent studies in the field.
- Figure S1 needs to improve. Bars (their colors) are unclear. I recommend to attach the legend similar to the fig 1C
- In line 336 word "conclusions" should be canceled
I recommend publication after minor revision.
Author Response
Response to Reviewer 2
We thank the reviewers for his/her constructive comments. In response to reviewer 2’s comments, we updated citation, modified the background section and updated the result to be more clearly presented.
We have also included a point-to-point response below to reviewers’ specific comments:
- The introduction seems to be too long. I recommend to transfer some parts to the discussion and compare literature data with results of the study in the discussion part. For example in lines 62-75 authors described own studies which correspond to the experiment from the manuscript
Response: As suggested by the reviewer, we have moved part of the introduction content to the discussion section (lines 352-359). This modification has further emphasized the relationship between neuroinflammation induced by sulfatide loss and metabolic phenotype we observed in current study.
- The small number of citations of works from the last 5 years is rather surprising (out of 50 bibliographic items, only 20 works have been published since 2018). Authors need to conduct detailed survey of the literature and supplement the citation with references to recent studies in the field.
Response: We appreciate reviewer for this comment. We did a detailed search among recent literature and updated the citation in our manuscript with more up-to-date publications. We also confirmed that all references are relevant to the contents of the manuscript. Currently, we have a total of 52 citations, 37 of which are publications from the past 5 years.
- Figure S1 needs to improve. Bars (their colors) are unclear. I recommend to attach the legend similar to the fig 1C
Response: We thank reviewer for this suggestion, we have included a figure legend for Figure S1 that is similar to Fig 1C.
- In line 336 word “conclusions” should be canceled.
Response: We appreciate reviewer for pointing this out, and we have deleted the word “conclusions” from line 336.
We hope that we have adequately addressed the questions identified in the previously submitted manuscript and that the revised manuscript will meet the reviewers’ approval. Thank you for your encouragement in resubmitting this manuscript.
Sincerely,
Xianlin Han, PhD
Professor of Medicine
Barshop Institute for Longevity and Aging Studies
Department of Medicine
UTHealth San Antonio
Phone: (210) 562-4104
Email: hanx@uthscsa.edu